# Dynamic Interplay of Metabolic and Transcriptional Responses in Shrimp during Early and Late Infection Stages of *Enterocytozoon hepatopenaei* (EHP)

**DOI:** 10.3390/ijms242316738

**Published:** 2023-11-25

**Authors:** Leiting Zhang, Sheng Zhang, Yi Qiao, Xiaohui Cao, Jie Cheng, Qingguo Meng, Hui Shen

**Affiliations:** 1Jiangsu Marine Fisheries Research Institute, Nantong 226007, China; 2College of Marine Science and Engineering, Nanjing Normal University, Nanjing 210023, China; 3School of Marine Science and Fisheries, Jiangsu Ocean University, Lianyungang 222005, China

**Keywords:** *Enterocytozoon hepatopenaei*, *Litopenaeus vannamei*, transcriptome, metabolomics, host-parasite interaction

## Abstract

*Enterocytozoon hepatopenaei* (EHP) is a microsporidian parasite that infects *Litopenaeus vannamei*, causing severe hepatopancreatic microsporidiosis (HPM) and resulting in significant economic losses. This study utilizes a combined analysis of transcriptomics and metabolomics to unveil the dynamic molecular interactions between EHP and its host, the Pacific white shrimp, during the early and late stages of infection. The results indicate distinct immunological, detoxification, and antioxidant responses in the early and late infection phases. During early EHP infection in shrimp, immune activation coincides with suppression of genes like Ftz-F1 and SEPs, potentially aiding parasitic evasion. In contrast, late infection shows a refined immune response with phagocytosis-enhancing down-regulation of Ftz-F1 and a resurgence in SEP expression. This phase is characterized by an up-regulated detoxification and antioxidant response, likely a defense against the accumulated effects of EHP, facilitating a stable host–pathogen relationship. In the later stages of infection, most immune responses return to baseline levels, while some immune genes remain active. The glutathione antioxidant system is suppressed early on but becomes activated in the later stages. This phenomenon could facilitate the early invasion of EHP while assisting the host in mitigating oxidative damage caused by late-stage infection. Notably, there are distinctive events in polyamine metabolism. Sustained up-regulation of spermidine synthase and concurrent reduction in spermine levels suggest a potential role of polyamines in EHP development. Throughout the infection process, significant differences in genes such as ATP synthase and hexokinase highlight the continuous influence on energy metabolism pathways. Additionally, growth-related pathways involving amino acids such as tryptophan, histidine, and taurine are disrupted early on, potentially contributing to the growth inhibition observed during the initial stages of infection. In summary, these findings elucidate the dynamic interplay between the host, *Litopenaeus vannamei*, and the parasite, EHP, during infection. Specific phase differences in immune responses, energy metabolism, and antioxidant processes underscore the intricate relationship between the host and the parasite. The disruption of polyamine metabolism offers a novel perspective in understanding the proliferation mechanisms of EHP. These discoveries significantly advance our comprehension of the pathogenic mechanisms of EHP and its interactions with the host.

## 1. Introduction

The Pacific white shrimp (*Litopenaeus vannamei*) has a pivotal role in the global shrimp aquaculture industry, contributing to over 70% of the total global shrimp production [1]. However, in recent years, the emergence of *Enterocytozoon hepatopenaei* (EHP) and its associated hepatopancreatic microsporidiosis (HPM) has led to significant economic losses in the Asian shrimp farming sector [2,3]. EHP is a single-celled eukaryotic microsporidian parasite belonging to the phylum microsporidia. It was initially identified in black tiger shrimp in Thailand in 2004 and was officially named based on histological, morphological, and phylogenetic data in 2009 [4,5,6]. Approximately half of the known microsporidia species can infect aquatic organisms [7]. Currently, three cultivated shrimp species are recognized as hosts for EHP: *P. monodon*, *L. vannamei*, and *P. stylirostris* [8]. While EHP infection does not cause lethal outcomes, it significantly suppresses shrimp growth rates and may induce immune dysfunction and complications such as White Feces Syndrome (WFS) [9,10]. Previous studies have indicated that EHP possesses a highly simplified genome and organelles, lacking metabolic pathways for glycolysis or oxidative phosphorylation to generate ATP [11]. Its genome is enriched with ATP/ADP transporters, indicating its reliance on continuously siphoning energy from the host [12]. Compared to other microsporidian parasites within the Enterocytozoonidae family, EHP exhibits unique patterns during host-parasite interactions [13].

The shrimp hepatopancreas possesses remarkable regenerative and self-repair mechanisms, mitigating some of the damage caused by EHP infection [14]. Consequently, EHP often establishes a long-term symbiotic relationship with shrimp [9]. Previous research has shown that shrimp with severe EHP infection exhibit down-regulated lipid metabolism compared to those with mild infection. Analysis of high (H) and low (L) infection severity groups revealed a positive correlation between infection severity and metabolic changes [13]. Additionally, as EHP infection intensifies, the abundance of certain microbial groups in shrimp intestines increases, including *Pseudomonas*, *Bradyrhizobium*, *Bacteroides*, and *Vibrio* [15]. These findings suggest that the effects on shrimp vary with the duration of EHP infection. Nevertheless, significant gaps remain in our understanding of the long-term effects of EHP infection in shrimp.

Transcriptome sequencing (RNA-Seq) enables the study of gene function under specific conditions through genomic structure analysis and functional annotation [16]. Metabolomics reveals changes in metabolic pathways through systematic analysis of small molecule metabolite spectra [17]. These omics techniques offer significant advantages, making them ideal for in-depth investigations into pathways and mechanisms of life events [18]. In recent years, multi-omics analyses have provided a comprehensive understanding of the molecular mechanisms of EHP infection. For instance, combined analysis of transcriptomics and the microbiome has revealed a positive correlation between *Bacillus* spp. in shrimp intestines and immune genes associated with antibacterial processes following EHP infection. Furthermore, integrated proteomic and metabolomic research has uncovered changes in growth-related proteins such as ecdysteroid-regulated protein and juvenile hormone esterase-like carboxylesterase 1, along with inhibited energy metabolism, contributing to the slow growth of shrimp following EHP infection [19,20]. 

In this study, through sampling of infected shrimp at two distinct life stages, we gained valuable insight into the EHP-shrimp interaction during different phases of the infection process. Comparative to single time-point analyses, analyzing juveniles versus adults offers a higher-resolution perspective into EHP pathogenesis during shrimp development. Our primary goal was to comprehensively investigate the interaction between Pacific white shrimp and EHP through a combined transcriptomics and metabolomics approach. By examining EHP infections in juvenile versus adult shrimp hepatopancreas, we explore the relationship between shrimp life stage and gene expression as well as metabolic pathways in diseased shrimp. Our research provides a rich dataset of the interaction between Pacific white shrimp and EHP, potentially yielding a better understanding of the changing impacts on shrimp during development and EHP progression, and offering deeper insights for future studies.

## 2. Results

### 2.1. Shrimp PCR Analysis and EHP Quantification Results

Our PCR analysis confirmed the presence of EHP in the infected groups and the absence of other pathogens across all samples. Specifically, adult shrimp infected with EHP (Asp group) exhibited a notably high EHP load, with an average of 2.6 × 10^6^ copies of EHP DNA per nanogram of host DNA. In contrast, juvenile shrimp infected with EHP (Jsp group) showed a lower EHP load, averaging 4.3 × 10^4^ copies/ng of host DNA. The uninfected adult (Asn group) and juvenile (Jsn group) shrimp showed no detectable levels of EHP, confirming their uninfected status (Figure 1). These results were consistent across three replicates for each group, ensuring the reliability of our pathogen load assessment.

### 2.2. Transcriptome Analysis of Hepatopancreas in Litopenaeus vannamei after EHP Infection

Twelve cDNA libraries were constructed from *Litopenaeus vannamei* hepatopancreas samples, with and without EHP infection. Sequencing on the Illumina platform yielded 110.15 Gb of high-quality clean data, with alignment ratios ranging from 78.37% to 91.91% to the NCBI reference genome. The clean data exhibited Q30 scores above 93.08%, GC content higher than 42%, and an error rate below 0.03%, ensuring its suitability for transcriptome analysis (Table 1).

After rigorous quality control to remove low-quality reads and short-length sequences, we obtained a total of 21,844 high-quality, non-redundant expressed unigenes. The annotation process, leveraging multiple public databases, yielded significant matches for the majority of these unigenes. Specifically, the NCBI non-redundant protein (Nr) database provided annotations for 96.75% of the genes, while the Cluster of Orthologous Groups (COG) database contributed to 69.36% of the gene annotations. Remarkably, a substantial portion of the unigenes, amounting to 96.86% (21,159 unigenes), were successfully annotated across all utilized databases, underscoring the comprehensiveness of our transcriptomic data (Table 2). Appendix A complements these findings by displaying the sequencing read distribution across the reference chromosomes, affirming the uniformity and depth of our sequencing coverage—a crucial factor for the assembly’s integrity and the overall quality of the transcriptome.

### 2.3. Sample Relationship Analysis

Principal coordinate analysis (PCoA) on the transcriptome data delineated distinct clustering of shrimp based on health status and developmental stage in the context of EHP infection (Figure 2). Notably, healthy juvenile shrimp (Jsn) and those at an early stage of EHP infection (Jsp) clustered more distinctly, indicating a more homogeneous transcriptomic profile within these groups. In contrast, the healthy adult shrimp (Asn) displayed a wider spread in the PCoA plot. Among the infected adult shrimp, Asp1 and Asp2 samples clustered closely, whereas Asp3 showed a marked departure along the PCoA2 axis. Interestingly, the PCoA plot positioned the Asn and Jsp groups in closer proximity, implying that despite the differences in developmental stage and infection status, there could be underlying transcriptomic similarities. Meanwhile, the clear demarcation between the other groups underscores the pronounced impact of EHP infection and developmental variation on the shrimp transcriptome.

### 2.4. Identification and Analysis of DEGs

Differential expression analysis was conducted using DESeq2. Unigenes were considered differentially expressed genes (DEGs) if they exhibited an absolute log2 fold change greater than 1 and an adjusted *p*-value below 0.05. The results are visually represented in Figure 3a–d through MA plots, delineating the DEGs between the infected and uninfected groups at distinct stages of infection.

In the comparison between the uninfected adult group (Asn) and the EHP-infected adult group (Asp), a total of 1769 DEGs were identified. Specifically, 963 genes were up-regulated and 806 genes were down-regulated in the Asp group relative to the Asn group (Figure 3a). Similarly, the comparison between the uninfected juvenile group (Jsn) and the EHP-infected juvenile group (Jsp) revealed 2444 DEGs, with 962 genes up-regulated and 1482 genes down-regulated in the Jsp group in comparison to the Jsn group (Figure 3b).

To discern the developmental differences between adults and juveniles, and to isolate the specific transcriptional changes attributable to EHP infection, we compared the uninfected groups (Asn vs. Jsn) and the infected groups (Asp vs. Jsp). In the Asn vs. Jsn comparison, 2687 DEGs were identified, indicating the inherent transcriptomic differences due to developmental stage (Figure 3c). This comparative approach allows us to distinguish between the transcriptional changes associated with normal development (Asn vs. Jsn) and those specifically induced by EHP infection (Asp vs. Jsp). By eliminating the background developmental differences, we aim to uncover unique DEGs that are specifically modulated due to the pathogenic effect of EHP. 

The Venn diagram in Appendix A illustrates 125 genes differentially expressed across all comparisons. Further examination of these 125 DEGs found that they exhibit opposite expression patterns between infected adults (Asp) and infected juveniles (Jsp) for most genes (Appendix A). The conflicting expression highlights significant interactions between infection status and developmental stage in the regulation of these genes. Therefore, these results reinforce the importance of analyzing adult and juvenile data separately, as adult and juvenile shrimp may utilize different genetic mechanisms to respond to EHP infection.

### 2.5. Functional Annotation Analysis of DEGs

Analysis of DEGs via KEGG functional annotation provides insight into the biological processes most impacted by EHP. In the late stage of infection (Figure 4a), adult shrimp exhibit a substantial number of DEGs in Metabolic pathways, particularly in carbohydrate and lipid metabolism. This indicates a major reprogramming of the host’s metabolic machinery, potentially to meet the energetic demands of an immune response or as a consequence of pathogen exploitation of host resources. In addition, signaling transduction pathways within the Genetic Information Processing (EIP) category are notably affected, alongside transport and catabolism within Cellular Processes (CP) and an overview of cancer-related genes within Human Diseases (HD).

In contrast, early-stage infected juvenile shrimp (Figure 4b) display a similar pattern of functional DEG annotation, yet with a higher number of DEGs, underscoring a more severe impact of EHP infection at this developmental phase, particularly concerning immune-related pathways (HD). Notably, the endocrine system within Organismal Systems (OS) is severely impacted, suggesting that hormonal dysregulation may contribute to the stunted growth observed in early-stage infected shrimp. Furthermore, signal transduction-related pathways, including translation in Genetic Information Processing (GIP) and signal transduction in EIP, are prominently affected, which could be associated with the pathogen’s early invasion mechanisms.

When comparing healthy adult shrimp with juvenile shrimp (Figure 4c,d), unique pathways emerge, such as signaling molecules and interaction, nervous system, and development and regeneration, which are absent in healthy comparisons. These pathways likely represent specific responses to EHP infection. The comparative analysis reveals that while there are inherent differences in the transcriptomic profiles of healthy adult and juvenile shrimp, EHP infection appears to diminish these distinctions across multiple categories, including HD, CP, OS, EIP, and Metabolism (M). This convergence could reflect a common infection response mechanism that overrides the transcriptomic differences normally present between adult and juvenile shrimp.

### 2.6. Pathway Enrichment Analysis of DEGs

To elucidate the biological functions of the DEGs, we carried out a comprehensive KEGG pathway enrichment analysis. The process involved the utilization of the Python Scipy (v1.11.1) software package, which facilitates the comparison of DEGs against the database of KEGG pathways to identify significant enrichments. We employed Fisher’s exact test to calculate the enrichment significance of each pathway.

To ensure the reliability of our findings, we adopted the Benjamini–Hochberg (BH) method to correct the multiple testing problem, thereby controlling the false discovery rate. We set a stringent threshold, where a Corrected *p*-value of less than 0.05 was considered indicative of significant enrichment. In our detailed examination of the transcriptomic data, Figure 5a illustrates the top 30 KEGG pathways that are significantly enriched in DEGs following EHP infection across different developmental stages of shrimp. These pathways provide insight into the stage-specific biological processes and molecular functions that are disrupted by the pathogen. Figure 5b zooms in on the pathways uniquely influenced by EHP infection, irrespective of the shrimp’s developmental stage. This analysis pinpoints the core molecular mechanisms consistently targeted by EHP across all infected individuals.

Our analysis revealed distinct metabolic responses between the infected adult and juvenile shrimp. In the Asn vs. Asp comparison, a significant enrichment of metabolic pathways was observed, with a notable focus on carbohydrate metabolism (Starch and sucrose metabolism), antioxidant processes (Ascorbate and aldarate metabolism), lipid metabolism (Sphingolipid metabolism), and amino acid metabolism (Glycine, serine, and threonine metabolism). Additionally, several pathways implicated in detoxification were also enriched. In contrast, the Jsn vs. Jsp comparison highlighted an enrichment of pathways associated with immune response and detoxification, including those involved in the metabolism of reactive oxygen species and xenobiotics by cytochrome P450. Pathways related to growth, such as steroid hormone biosynthesis, were also more prominent in juvenile shrimp, suggesting an early-stage toxic response to EHP infection that influences immunity, detoxification, and hormone regulation. Adult shrimp, on the other hand, showed signs of metabolic adaptation to EHP infection in later stages, with a focus on energy metabolism. Nonetheless, the persistence of enriched detoxification pathways indicates ongoing toxic challenges due to EHP. The specific distribution of DEGs across these pathways is detailed in Appendix A, providing a comprehensive view of the transcriptional changes associated with EHP infection at different developmental stages.

### 2.7. Validation of Transcriptome Data by qRT-PCR

Gene expression levels of 24 genes were assessed via qPCR. The obtained data revealed that the expression trends observed in qPCR and RNA-Seq analyses exhibited a concordant pattern, as depicted in Appendix A. These results support the dependability of transcriptome high-throughput sequencing outcomes.

### 2.8. QC Metabolome Sample Principal Component Analysis

This study employed untargeted metabolomics analysis to investigate metabolites in the hepatopancreas of Pacific white shrimp at different time points after EHP infection, aiming to identify metabolites potentially related to infection duration. Principal Component Analysis (PCA) results demonstrated satisfactory sample classification in both positive and negative ion modes (Figure 6a). Partial Least Squares Discriminant Analysis (PLS-DA) indicated differences between healthy adults and juveniles (Asn and Jsn), while post-EHP infection, the two infected groups (Asp and Jsp) exhibited some similarity (Figure 6b). Furthermore, evaluation parameters for the PLS-DA model were as follows: for positive ion mode, R2X(cum) = 0.516, R2Y(cum) = 0.62, Q2 = 0.368; for negative ion mode, R2X(cum) = 0.549, R2Y = 0.598, Q2 = 0.336, suggesting model stability and reliability without overfitting (Figure 6c).

In our investigation of Selectivity Differential Metabolites (SDMs), Orthogonal Partial Least Squares Discriminant Analysis (OPLS-DA) served as the basis for screening. We identified SDMs based on the criteria of a *p*-value below 0.05, a Variable Importance in Projection (VIP) score exceeding 1, and an absolute fold change (|FC|) greater than 1. The volcano plot analyses, presented in Figure 7a–d, illustrate distinct shifts in metabolite regulation under various conditions.

In the Asn vs. Asp comparison, we identified 143 metabolites that were up-regulated and 180 that were down-regulated in the infected adult shrimp (Asp) relative to the control adult group (Asn). For the juvenile shrimp, the Jsn vs. Jsp comparison revealed 108 metabolites up-regulated and 266 down-regulated in the infected juveniles (Jsp) compared with the uninfected juveniles (Jsn). When comparing the adult and juvenile uninfected shrimp (Asn vs. Jsn), there were 278 metabolites up-regulated in the adults (Asn) alongside 82 down-regulated metabolites. The contrast between the infected adult and juvenile shrimp (Asp vs. Jsp) showed 68 metabolites up-regulated in the adult infected group (Asp) and 49 down-regulated.

Taking into account all four comparison pairs—Asn vs. Asp, Jsn vs. Jsp, Asn vs. Jsn, Asp vs. Jsp—a total of 323, 374, 360, and 117 SDMs were identified, respectively. Notably, we discovered seven metabolites consistently modulated across all comparison groups, indicative of a core metabolic response to EHP infection and developmental stage differences (highlighted in Figure 7e). These shared SDMs, consistently dysregulated in each condition, may provide valuable insights into the metabolic pathways universally affected by EHP infection across different life stages of the shrimp.

### 2.9. Identification of Significantly Different Metabolites and KEGG Enrichment Analysis

To further explore potential metabolic pathway alterations in the hepatopancreas of shrimp at different time points after EHP infection, we conducted additional analysis on the SDMs using the KEGG database. KEGG pathway enrichment analysis revealed that in the early response to EHP infection, the purine, pyrimidine, histidine, tryptophan, alpha-linolenic acid, taurine, and hypotaurine metabolism pathways, as well as glycosaminoglycan biosynthesis, were significantly enriched (Figure 8b). In contrast, the pyrimidine, glycerophospholipid, beta-alanine, tryptophan, sphingolipid, arginine, and proline metabolism pathways, as well as lysine biosynthesis, were significantly enriched in the later stages of EHP infection (Figure 8a). These results suggest that in the early stages of infection, EHP induces significant disruptions in amino acid metabolism, particularly affecting essential amino acids for growth like tryptophan and histidine. In the later stages, EHP primarily disrupts lipid metabolism and energy production pathways in mature shrimp. Enrichment analysis also indicated significant enrichment of the sphingolipid, histidine, phenylalanine, alpha-linolenic acid, and tryptophan metabolism pathways, as well as Arginine biosynthesis and lysine biosynthesis in the Asn vs. Jsn comparison (Figure 8c). Tryptophan, Phenylalanine, and glycerophospholipid metabolism pathways, along with lysine degradation and Pentose and glucuronate interconversions, were significantly enriched in the Asp vs. Jsp comparison (Figure 8d).

### 2.10. Cluster Analysis of Differential Metabolites

To investigate the temporal dynamics of metabolite changes in the hepato-pancreas of shrimp infected with EHP, this study utilized the Mufzz package in R for cluster analysis. This enabled a detailed comparison of metabolite abundances between juvenile and adult shrimp. Within the clustering results, Cluster 3 represents metabolites (46 SDMs) with increasing abundance from the uninfected group to the long-term infected group. Key pathways mapped by these metabolites include tryptophan, caffeine, pyrimidine, purine, and glycerophospholipid metabolism pathways, along with lysine degradation (Figure 9a). The sustained accumulation of essential amino acids might suggest that under the influence of EHP, shrimp engage in compensatory growth strategies. The continuous increase in glycerophospholipid-related metabolites might reflect EHP’s specific requirement for phospholipids in lipid metabolism. On the other hand, Clusters 5 and 8 represent metabolites (182 SDMs) with decreasing abundance from the uninfected group to the long-term infected group. Key pathways involve beta-alanine, glutathione, tryptophan, sphingolipid, taurine, and hypotaurine metabolism pathways, along with glycosaminoglycan biosynthesis (Figure 9b). The persistent down-regulation of metabolites like taurine and sphingolipids might indicate ongoing and progressively severe disruptions to shrimp’s metabolism and lipid metabolism. Notably, we observed in the transcriptome analysis that genes related to glutathione were suppressed in the early infection group (Jsp) and significantly activated in the late infection group (Asp). However, the metabolism of glutathione continued to decrease. This suggests that glutathione might be an important targeted metabolite concerning EHP growth and its interaction with the host. Lastly, Clusters 6 and 7 represent metabolites (133 SDMs) that change in abundance during the initial EHP infection and show a certain recovery trend as infection time increases. These metabolites are involved in pyrimidine, arachidonic acid, purine, cysteine, and methionine metabolism pathways, along with lysine degradation (Figure 9c). The recovery of pyrimidine and purine metabolism might indicate the restoration of some energy metabolism and a reduction in inflammation responses within the shrimp. These metabolites could serve as potential indicators of adaptive responses to infection. Among them, lysine degradation seems to be significantly affected, enriched in Clusters 3, 6, and 7. This might suggest that lysine is an important amino acid for EHP growth and a critical factor in the interaction between EHP and the host.

### 2.11. Transcriptome and Metabolome Correlation Analysis

Our integrative analysis, depicted in Appendix A, revealed a total of 21 co-expressed pathways that were highly enriched, signifying their potential role in the host response to EHP infection. Detailed in Appendix A, shows all DEGs analyzed in the conjoint analysis. Notably, during the early stages of infection, purine metabolism and glutathione metabolism emerged as critical pathways. These pathways are implicated in nucleotide turnover and redox homeostasis, respectively, which are vital processes during the cellular response to pathogenic stress. In the later stages of infection, our analysis identified three predominant pathways: Glycerophospholipid metabolism, sphingolipid metabolism, and glutathione metabolism. The enrichment of these pathways underscores their importance in membrane integrity, signaling, and again, redox balance, which may be pivotal for the shrimp’s ability to cope with the persistent effects of EHP. The convergence of glutathione metabolism across both early and late stages of infection highlights its central role in the shrimp’s defense mechanism against the oxidative stress induced by EHP.

#### 2.11.1. Effect of EHP on Purine Metabolism during Early Infection

During the early stages of shrimp infection with EHP, 8 SDMs and 25 DEGs were involved in purine metabolism (Figure 10). In the degradation of adenosine triphosphate (ATP) and guanosine ribonucleoside (GuaR), numerous up-regulated DEGs were observed. Conversely, in the oxidation pathway of xanthine, many down-regulated DEGs were observed, accompanied by significant accumulation of uric acid and hypoxanthine in this pathway. Additionally, significant accumulations of Deoxyinosine and Xanthosine, both related to ABC transporters, were noted, along with significant accumulations of other purine metabolism-related metabolites, including ADP, Guanosine diphosphate, and IDP. Notably, the activation of numerous transport proteins and higher energy metabolism might suggest a strong targeting effect of EHP on host energy metabolism during the early infection. The activation of more purine nucleotide metabolism-related processes indicates that EHP might rely on host ATP and GuaR to complete its own DNA and RNA biosynthesis. On the other hand, most associated genes exhibited significant up-regulation at the transcriptional level, such as ADP-sugar pyrophosphatase, guanine deaminase, AMP deaminase 2, and purine nucleoside phosphorylase. Meanwhile, a considerable reduction in the transcriptional level of xanthine dehydrogenase (XDH) was observed, which would lead to the significant accumulation of Xanthosine and Hypoxanthine within shrimp, indicating ineffective metabolism of these compounds.

#### 2.11.2. Effect of EHP on Glutathione Metabolism during Early Infection

During the early stages of shrimp infection with EHP, 1 SDM, and 43 DEGs were enriched in the glutathione metabolism pathway (Figure 11). Among these, the majority of DEGs were significantly down-regulated, suggesting that the host’s glutathione metabolism was suppressed during the early infection. This result might indicate an initial stress response of shrimp related to glutathione. Interestingly, we observed that some DEGs involved in polyamine metabolism were significantly up-regulated, while the abundance of putrescine was significantly down-regulated. Furthermore, we found a significant up-regulation of the downstream gene ornithine decarboxylase (2.5.1.22) related to spermine synthesis, which could be one of the reasons for the significant decrease in putrescine abundance. However, we did not observe an up-regulation in spermine abundance. This might suggest EHP’s reliance on polyamine metabolism and its targeted effect on the host’s polyamine metabolism pathways.

#### 2.11.3. Effect of EHP on the Metabolism of Glycerophospholipids and Sphingolipids during Late Infection

In the late stages of shrimp infection with EHP, we observed a significant impact on glycerophospholipid metabolism, with 9 SDMs and 9 DEGs enriched in this pathway (Figure 12). Among these, six significantly accumulated SDMs were annotated as 1-Acyl-sn-glycero-3-phosphocholine, and the remaining accumulated SDMs were Acetylcholine, Phosphatidyl ethanolamine, and Phosphatidyl serine. Changes in the levels of these glycerophospholipid metabolites could lead to alterations in cell membrane composition, affecting the function and characteristics of cell membranes. Additionally, these changes in SDMs could be crucial for EHP’s invasion of host cells. Several DEGs in this pathway exhibited significant up-regulation, mostly functioning as upstream genes for the aforementioned SDMs. These included lysophospholipid acyltransferase, ethanolamine kinase, and lysophosphatidylcholine acyltransferase. Although we did not find significantly down-regulated metabolites in this pathway, most DEGs in this pathway showed significant down-regulation at the transcriptional level. These DEGs included phosphatidylserine synthase, glycerol-3-phosphate O-acyltransferase, ethanolamine-phosphate cytidylyltransferase, ethanolamine-phosphate phospho-lyase, phosphoethanolamine N-methyltransferase, and lysophospholipase I. Notably, the down-regulation of phosphoethanolamine N-methyltransferase was significant, which could hinder phosphatidylcholine formation and lead to the accumulation of phosphatidyl ethanolamine. This might be an important targeted metabolite by EHP during the later stages of infection.

In lipid metabolism, we also observed a significant impact on sphingolipid metabolism during the later stages of infection, with 2 SDMs and 20 DEGs enriched in this pathway (Figure 12). Among these, we found a significant down-regulation of sphingomyelin phosphodiesterase, which could contribute to the significant decrease in Psychosine abundance. This decrease in Psychosine could disrupt primarily sphingolipid-based signal transduction. Additionally, we observed a significant decrease in sphingosine 1-phosphate abundance. Interestingly, sphinganine-1-phosphate aldolase was up-regulated. This might indicate that in the later stages of infection, the host might compensate for the transitional consumption of sphingosine 1-phosphate. This also suggests a potential association between sphingosine 1-phosphate and EHP’s growth.

#### 2.11.4. Effect of EHP on Glutathione Metabolism during Late Infection

In this study, we discovered a striking contrast between the late and early stages of infection in the glutathione metabolism pathway, with 1 SDM and 20 DEGs enriched (Figure 13). We observed a significant up-regulation of numerous DEGs, with the majority of them being glutathione S-transferase. This indicates that the shrimp’s detoxification and antioxidant defense functions are gradually recovering or even becoming more significantly activated. Notably, we also found a significant up-regulation of spermine synthase in the later stages of infection. Moreover, this up-regulation was more pronounced compared to the early stages of infection. However, the abundance of spermidine was not only significantly down-regulated but also lower compared to the early stages of infection. These findings further support the potentially important role of polyamines like spermine and spermidine in EHP’s growth and development. The opposing regulation of this pathway in the early and late stages highlights the significance of the glutathione metabolism pathway as a key interaction point between EHP and the host.

## 3. Discussion

Microsporidian infections often impact the host’s behavior and interact with various regulatory mechanisms within the host’s body [21]. In this study, for the first time, we revealed distinct patterns of host response through infection samples taken at different time points, observing significant differences between early and late infection stages. We identified 2444 DEGs and 374 SDMs during the early infection, and 1769 DEGs and 323 SDMs during the late infection. PCA analysis demonstrated substantial differences in transcription and metabolic levels between juveniles and adults in the absence of EHP infection. However, these differences diminished after EHP infection, with partial similarity observed between the two groups. These findings suggest the activation of immune defense-related genes in both shrimp groups. The early immune response appeared to be broad spectrum, yet a substantial portion of immune-related genes remained suppressed, possibly due to EHP invasion. In the late infection stage, most immune responses gradually recovered, and some immune-related genes exhibited more pronounced activation, likely indicating the shrimp’s adaptation to EHP infection. Moreover, both shrimp groups experienced profound impacts on metabolic pathways. Energy metabolism pathways were significantly enriched in both early and late stages, highlighting the potential importance of energy metabolism pathways for EHP at both time points, consistent with EHP’s absolute dependence on host energy [22]. Additionally, we identified certain energy metabolism-related genes as potential pathways for EHP to acquire energy. Notably, immune response-related DEGs were significantly affected during the early infection stage, with a large number of up-regulated DEGs indicating EHP activation of host defense mechanisms. However, down-regulated expression of certain DEGs may signify key pathways for EHP invasion into host cells and interaction with the host. Importantly, we also observed disruptions in growth-related metabolic pathways, likely contributing to the slow growth of juveniles following EHP infection. In the late infection stage, EHP primarily disrupted lipid, energy, and carbohydrate metabolism pathways, likely due to the extensive reproduction of spores within shrimp, which divert host energy and resources to sustain their proliferation. Appendix A details the expression profiles of select DEGs across different developmental stages, corroborating our findings.

Under the stress of EHP infection, there are significant differences in the immune responses of shrimp during the early and late stages. Shrimp identified more immune-related DEGs during the early infection compared to the late infection, suggesting that EHP may activate a broad spectrum of immune defenses in the early stages. In the late infection stage, the expression of these DEGs returned to normal, possibly indicating a strategy employed by EHP to establish a long-term symbiotic relationship with the shrimp while reducing the host stress response. However, during the early infection, we observed that certain immune-related genes might be regulated by EHP to facilitate invasion. Previous research indicates that microsporidia can suppress host immunity during invasion [23].

We observed a significant impact on the transcriptional expression of two genes, Ftz-F1 and SEPs, during the early infection stage. Ftz-F1 overexpression can inhibit phagocytosis of blood cells. This suggests that in the early stages of EHP infection, the parasite might manipulate the host’s Ftz-F1 to evade phagocytosis and promote its infection [24,25]. Similarly, it is known that small open reading frame (sORF)-encoded peptides (SEPs) can inhibit the proliferation of white spot syndrome virus (WSSV) [26]. The down-regulation of SEPs-related genes during the early stages of EHP infection might lead to a decrease in antimicrobial peptide (AMP) levels, promoting EHP invasion. Additionally, in the Asp vs. Jsp comparison, we identified a unique significant enrichment of the PI3K-Akt signaling pathway. The PI3K-Akt pathway is involved in shrimp’s immune response and plays a crucial role in shrimp’s defense against bacteria and viruses [27,28]. This suggests that PI3K-Akt may also play an important role in EHP invasion. In summary, EHP significantly impacts the host’s immune system during the early stages of infection.

In contrast, the reduced number of DEGs during the late infection stage suggests that the effects of EHP are gradually diminishing. This includes most immune-related DEGs that exhibited significant differences in the early stages, such as the expression of SEPs which returned to normal in the late infection stage. Interestingly, some immune activities become more pronounced. For example, the expression of Ftz-F1 is significantly down-regulated in the late infection stage, leading to a significant increase in the cell’s phagocytic ability against pathogens [24]. Additionally, we observed that antioxidant and detoxification-related pathways, such as oxidative phosphorylation and cytochrome P450 xenobiotic metabolism pathways, seem to play an important role in shrimp during the late infection stage. This includes the significant activation of pathways related to ascorbic acid, aldehyde metabolism, and detoxification. Ascorbic acid (vitamin C) is generally considered a potent antioxidant capable of scavenging free radicals [29]. However, recent studies have shown that high doses of vitamin C can also have pro-oxidant effects, protecting larvae of Daphnia magna from *Vibrio harveyi* infection [30]. We observed that the expression of ascorbic acid is significantly down-regulated during the early infection stage but significantly up-regulated during the late infection stage. Taken together, these results suggest that in the late infection stage, the high parasite load of EHP might generate harmful substances to the host. To counteract the toxic effects of these substances, shrimp may protect themselves by continuously up-regulating detoxification and antioxidant systems. Additionally, to cope with prolonged infection, the shrimp’s immune system gradually recovers and remains in a continuously activated state. In conclusion, these strategies may explain why shrimp can establish a long-term symbiotic relationship with EHP during the later stages of infection.

In the context of metabolic pathways, our data indicate a strong reliance of EHP on the host’s energy metabolism. During the early infection stage, ATP synthase exhibits significant up-regulation. ATP synthase is essential for electron transport and ATP synthesis [31]. Moreover, other studies have shown that white spot syndrome virus (WSSV) up-regulates ATP synthase in infected blood cells to enhance its own replication [32]. Considering that EHP is nearly incapable of ATP production and relies entirely on obtaining ATP from host cells [13], we speculate that EHP might similarly up-regulate the expression of ATP synthase in host cells and heavily exploit the host’s ATP to support its proliferation during the early infection stages. Interestingly, within the purine metabolism pathway, we also observe significant up-regulation of genes involved in the degradation of ATP and GuaR (Figure 9). ATP and GuaR are vital components of the purine metabolism pathway, playing crucial roles in energy transfer and nucleic acid synthesis. Previous research indicates that microsporidia lack many of the genes required for nucleotide synthesis. However, they retain the core enzyme toolkit for nucleotide salvage, suggesting that microsporidia manipulate host cell processes to promote nucleotide synthesis and hijack the host’s ATP and nucleotides [33]. Additionally, radiolabeled purine nucleotides have been utilized by *Trachipleistophora hominis* for the critical purine building blocks of its DNA and RNA [34]. Our study suggests that EHP might manipulate host nucleotide-regulating genes to facilitate its replication. The up-regulation of genes involved in the degradation of ATP and GuaR further supports this notion. Additionally, we observe a significant enrichment of cytochrome c oxidase (COX) during the early infection stage. Previous research suggests that microsporidia can gain energy by interacting with the host mitochondria through spore surface protein (rEhSSP1) [35]. COX is mainly found in the inner mitochondrial membrane, catalyzing redox reactions [36]. Mitochondria, as the main energy-producing organelles, may be a prime target for EHP. Therefore, we speculate that COX might be an essential pathway for EHP to acquire mitochondrial energy. However, further research is required to confirm this hypothesis and elucidate the exact mechanism through which EHP exploits host nucleotide resources.

In contrast, during the late infection stage, EHP seems to become more reliant on the host’s energy metabolism. Several fundamental ATP-generating pathways, including glycolysis and oxidative phosphorylation, are significantly affected. Glycolysis is a key carbohydrate metabolism pathway. Although genes related to glycolysis are missing in the Enterocytozoon family of microsporidia, each lineage retains genes from different parts of the glycolytic pathway [13]. This suggests that glycolysis remains an important energy-generating pathway in these microsporidia. Phosphoglucomutase (PGM) catalyzes the interconversion of glucose 1-phosphate and glucose 6-phosphate. This allows microsporidia to use glucose 1-phosphate in glycolysis and glucose 6-phosphate (G6P) in trehalose synthesis [37]. Our data indicate a significant up-regulation of PGM during the late infection stage. Hexokinase, which is important for glycolysis, is significantly down-regulated in the Asn and Asp groups. EHP possesses only one copy of hexokinase and lacks a complete glycolytic pathway. Yet, it utilizes secreted hexokinase to disrupt host metabolism and support its development, similar to other Enterocytozoon microsporidia [38,39,40,41]. Notably, hexokinase, as the enzyme responsible for the first step of glycolysis, also catalyzes the conversion of G6P. Furthermore, G6P might promote increased ATP production for microsporidian growth in the host. A significant up-regulation of G6P has also been observed after *Nosema bombycis* infection, another microsporidian [42,43]. These findings suggest that PGM, hexokinase, and G6P play a crucial role in EHP’s energy acquisition pathway. EHP may target the host’s PGM to acquire the necessary G6P, while the down-regulation of hexokinase might indicate the parasite’s manipulation of the host’s enzymes to complete its glycolysis. In conclusion, these enzymes and G6P likely play key roles in obtaining ATP, regulating the host, and promoting EHP’s lifecycle, especially during the progression of infection.

Lipids, as a principal component of hepatopancreas, play a vital role in energy storage, among other functions. During the late infection stage, the high load of EHP leads to the disruption of hepatopancreas function, resulting in the disturbance of lipid metabolism pathways. In this study, we observed a significant enrichment in glycerophospholipid and sphingolipid metabolism pathways. This aligns with Ding et al.’s research, which showed that microsporidian infection disrupts glycerophospholipid and sphingolipid metabolism in *Eriocheir sinensis* [44]. Our temporal analysis further corroborated this transition, indicating a sustained increase in glycerophospholipid-related metabolism from the non-infected group to the long-term infected group. Glycerophospholipids, also known as phospholipids, are implicated in the synthesis of membrane phospholipids and possess crucial roles in various microsporidia for membrane synthesis and other cellular functions [45]. The significant accumulation of phospholipids suggests that EHP may have specific requirements for phospholipids during its growth and development. Furthermore, we observed a significant down-regulation of phosphoethanolamine N-methyltransferase. El Alaoui et al. suggested the presence of phospholipid synthesis pathways in microsporidia and observed elevated activities of phospholipid acylserine decarboxylase and phospholipid ethanolamine N-methyltransferase following microsporidian infection. The combined action of these enzymes can convert phosphatidylethanolamine and phosphatidylserine into phosphatidylcholine [46]. Additionally, another study demonstrated that phosphatidic acid is a limiting host metabolite for *Tubulinosema ratisbonensis* proliferation [47]. These findings indicate that the enrichment of phospholipids might signify EHP’s regulation of PE and PS-related genes, with phospholipids potentially playing a significant role in EHP’s growth and development.

Sphingolipids, as another important class of lipids, are integral components of membrane lipids and play vital roles in signal transduction. In this study, we observed a continuous decrease in sphingolipids as the infection cycle progressed. Existing research suggests that microsporidia might form invasion vesicles through interactions with the host cell membrane during invasion [44,48]. This implies that EHP might utilize potential interactions with the host cell membrane to facilitate invasion. Sphingomyelin can be transformed into sphingosine through the action of serine palmitoyltransferase long chain (SPTL). Sphingosine can further react to form ceramide. The work of JH Jeon et al. discovered that sphingolipid levels might impact microsporidian proliferation. This is because microsporidia might have diverse enzymes generating C20-ceramides, and the study indicated that only the infected group exhibited an enrichment of ceramide species, suggesting that microsporidia could hijack host enzymes to generate ceramides. This indicates that sphingolipids, especially sphingosine, might play a significant role in microsporidian infection, potentially promoting their growth [49]. Our data display a notable reduction in sphingosine and a significant up-regulation of ceramide phosphate ethanolamine lyase, which might result in an overall decrease in sphingosine levels. This suggests that EHP might similarly intervene in host sphingolipid metabolism and promote its proliferation. In conclusion, these lipid changes pose intriguing questions about their impact on the EHP-host interaction and the mechanisms through which EHP manipulates host lipid metabolism, warranting further investigation.

Our study also revealed a significant down-regulation of genes involved in glutathione metabolism during the early EHP infection period. The glutathione system plays a crucial role in antioxidant defense and maintaining redox balance, which are essential during the infection process. A study focusing on *Nosema ceranae* highlighted the importance of the glutathione system in microsporidian invasion. Knocking down the expression of γ-glutamyl-cysteine synthetase and thioredoxin reductase genes resulted in a significant reduction in spore load within honeybees [50]. Interestingly, our data indicate a substantial inhibition of glutathione metabolism during the early infection phase, which becomes significantly activated in the later stages of infection. This suggests that the glutathione system may serve as a crucial defense mechanism against EHP invasion in the early stage, but it is significantly suppressed in the context of EHP’s invasion strategy. In the later stages, the substantial spore load might prompt the activation of shrimp’s antioxidant defense to cope with prolonged microsporidian infection. However, the exact role of the glutathione system in the EHP-host interaction, as well as how EHP suppresses the host’s glutathione system to facilitate invasion in the early phase, remains to be elucidated.

It is worth noting that polyamine metabolism, as a part of the glutathione metabolic pathway, is significantly affected both in the early and later infection phases. Polyamines include putrescine, spermidine, and spermine. Notably, spermidine levels show significant down-regulation in both early and late infection phases, while the expression of spermine synthase is significantly up-regulated in both periods. Interestingly, the reduction in spermidine levels is more pronounced in the later phase, while the transcriptional level of spermine synthase is even higher in the later phase. Polyamine metabolism is considered important for parasite survival [51]. Related studies have indicated that *Enterocytozoon cuniculi* exhibit polyamine synthesis and transformation activities before germination, shown by the conversion of spermine to spermidine. Furthermore, the use of polyamine analogs as inhibitors effectively interferes with spore uptake and conversion of polyamines before germination [52]. This suggests that polyamines may play a crucial role in microsporidian survival, particularly in relation to germination activity. However, further research suggests that microsporidia have adapted to rely more on polyamine uptake and interconversion rather than de novo synthesis [53]. Some microsporidia, like *Enterocytozoon cuniculi*, demonstrate significant potential for polyamine transformation [52]. These findings indicate that polyamines are vital for microsporidian survival and proliferation, with uptake being more critical than synthesis. This seems to explain the continuous up-regulation of spermine synthase expression and the persistent down-regulation of spermidine levels observed in our study. The role of polyamines in EHP infection and how EHP regulates host polyamine metabolism merits further investigation. Additionally, polyamine analogs could hold potential research value in EHP control strategies.

Finally, in the infection cycle of EHP, we observed that shrimp seem to experience more pronounced growth inhibition during the early infection phase. In the early stages of EHP infection, we observed significant disruption in amino acid metabolism, particularly for growing essential amino acids such as tryptophan and histidine. Our KEGG topology analysis of SDMs suggests that this disruption could impair shrimp growth and development, as these amino acids are crucial for protein synthesis and various metabolic processes [54] Furthermore, the enrichment of pathways related to purine, pyrimidine, α-linolenic acid, taurine, and hypotaurine metabolism, as well as glycosaminoglycan biosynthesis, indicates that EHP might interfere with host nucleotide metabolism, fatty acid metabolism, osmotic regulation, and extracellular matrix composition during the early infection phase. Our temporal analysis also revealed sustained down-regulation of taurine metabolism. Another microsporidian, *Paranosema locustae*, significantly reduces taurine levels in infected *Locusta migratoria*, leading to growth inhibition in the infected locusts [55] This suggests that the persistent down-regulation of taurine could be another important factor contributing to the growth inhibition and molting impairment in shrimp after EHP infection. Moreover, Li et al.’s study showed that when taurine was supplemented to infected locusts through injection, the delayed development phenomenon was suppressed [55] However, it is worth noting that taurine metabolism mechanisms in shrimp and locusts could be different, and the specific impact of EHP infection on shrimp taurine metabolism requires further investigation.

In our study, we also found that certain metabolic pathways, such as tryptophan, phenylalanine metabolism pathways, and lysine degradation pathways, were significantly enriched in both early and late infection stages, albeit in different host groups (Asn vs. Jsn and Asp vs. Jsp). Among them, lysine degradation seems to be more affected, as it was significantly enriched in Clusters 3, 6, and 7 in the temporal analysis. Lysine-rich proteins, such as extreme-heat pipe protein 2 (EHPPTP2), were identified in EHP’s infective polar tube [56] This suggests that lysine could be an important amino acid for EHP growth and host interaction, possibly contributing to the structure, function of EHPPTP2, and the mechanisms of EHP infection. In conclusion, the early growth inhibition observed in shrimp might be due to the widespread effects caused by EHP, with inhibition of amino acids and steroid metabolism closely associated with growth possibly leading to stunted growth in shrimp.

In summary, our study provides new insights into the metabolic adaptations of shrimp during EHP infection. However, our understanding of the complex interactions between EHP and its host is still in its early stages. Further research is needed to validate these findings and uncover the molecular mechanisms behind these metabolic changes. This study not only enhances our understanding of the pathogenesis of EHP but also identifies potential targets for therapeutic intervention.

## 4. Materials and Methods

### 4.1. Pacific White Shrimp Samples

In August 2022, we identified EHP infection in samples collected from ponds located in Dafeng District, Yancheng City, Jiangsu Province, using EHP PCR. Diseased shrimp samples were collected from the ponds, measuring approximately 3.9 ± 0.2 cm in size. Simultaneously, healthy shrimp under identical conditions were also collected, measuring approximately 4.6 ± 0.2 cm. Thirty days later, we collected samples from the same batch of diseased and healthy shrimp in the previously mentioned ponds. The size of the diseased shrimp was approximately 8.1 ± 0.2 cm, while the healthy shrimp measured approximately 11.3 ± 0.2 cm. After collection, the shrimp were individually reared for subsequent pathogen detection and RNA extraction. The rearing system maintained a constant temperature of 24 ± 1 °C and a salinity of 12 ± 2 PPT.

Before RNA extraction, four groups of samples were tested for pathogen PCR and EHP content to confirm that there was no other pathogen infection and to determine the degree of EHP infection. The four groups were EHP-negative juvenile shrimp (Jsn), EHP-positive juvenile shrimp (Jsp), EHP-negative adult shrimp (Asn), and EHP-positive adult shrimp (Asp). This was performed to confirm the absence of other pathogen infections and to assess the extent of EHP infection. To minimize individual variations, three shrimp liver and pancreatic mixtures were combined to form one sample for each group. For transcriptome and metabolomics analysis, three replicate samples were taken from each group. All samples were immediately frozen in liquid nitrogen and stored until RNA and metabolite extraction.

### 4.2. Multiple Pathogen Screening and Quantitative PCR Test of EHP

For the detection of pathogens in our samples, we followed the established PCR protocols as described by Shen et al. [15]. This involved screening for a panel of shrimp pathogens, including white spot syndrome virus (WSSV), yellow head virus (YHV), acute hepatopancreatic necrosis disease (AHPND), infectious hypodermal and hematopoietic necrosis virus (IHHNV), Taura syndrome virus (TSV), and EHP. The presence of these pathogens was determined by analyzing the PCR amplification products via agarose gel electrophoresis.

To quantify the EHP load in infected shrimp, we employed TaqMan probe real-time PCR, a more sensitive and specific method that allows for the precise measurement of pathogen copies [57]. We used a plasmid standard containing the EHP SSU rDNA gene, which was serially diluted to generate a standard curve for quantification. Each sample was run in triplicate to ensure accuracy, and the number of EHP copies per nanogram of hepatopancreatic (HP) DNA was calculated by normalizing the EHP copy number to the HP DNA quantity. The detailed protocol for the TaqMan probe real-time PCR is provided in Appendix A.

### 4.3. RNA Extraction and Quality Control

Total RNA was extracted from the samples using TRIzol^®^ Reagent (Invitrogen, Carlsbad, CA, USA) as per the manufacturer’s instructions. To assess the concentration, purity, and integrity of the extracted RNA, Nanodrop2000, agarose gel electrophoresis, and Agilent2100 were employed [58]. High-quality RNA samples (OD260/OD280 = 1.8~2.2, OD260/OD230 ≥ 2.0, RIN ≥ 8.0, 28S:18S ≥ 1.0, >1 μg) were then used to construct sequencing libraries.

### 4.4. Assembly of Transcriptomic Data

After extracting RNA, we proceeded to construct the mRNA library for sequencing. mRNA was isolated from the total RNA using magnetic beads coated with oligo(dT) to selectively bind the polyA tail of mRNA molecules. The isolated mRNA was then fragmented using a fragmentation buffer under optimized conditions. This was followed by the synthesis of first-strand cDNA using reverse transcriptase with the fragmented mRNA as a template, and subsequently, second-strand cDNA was synthesized to form a stable double-stranded structure using random primers, as described in [59]. The cDNA fragments underwent an end repair process, and a single ‘A’ nucleotide was added to the 3’ ends to facilitate adapter ligation of the Y-shaped adapters necessary for sequencing. After ligation, the library was purified and size-selected to ensure uniformity in fragment length. PCR amplification of the size-selected fragments was then performed, and the PCR products were subjected to a final purification step to obtain the completed library. Sequencing was carried out on the Illumina NovaSeq 6000 platform using the NovaSeq Reagent Kit, which generated the mapped reads necessary for subsequent transcript assembly and quantification of expression levels. Detailed protocols and parameters for library construction and sequencing are provided to ensure the reproducibility and transparency of our methodology.

### 4.5. Processing and Annotation of Transcriptome Data

In our bioinformatics pipeline, we initially aligned the RNA sequencing reads to the EHP reference genome (GCA_002081675.1) using HiSat2 (v2.1.0) to identify and separate EHP-specific sequences. Reads that did not align with the EHP genome were extracted using Samtools (v1.9) and converted into fastq files for subsequent analysis. These unaligned reads were then aligned to the *Litopenaeus vannamei* reference genome (GCF_003789085.1) using HiSat2, ensuring that our transcriptome data mapping was specific to the shrimp host and free from EHP contamination. Following alignment, we employed StringTie (v2.1.2) for de novo assembly and gene annotation of the shrimp transcriptome. The quality of the assembled transcripts was rigorously evaluated using established benchmarks [60]. For functional annotation, the transcripts were cross-referenced against multiple databases, including Nr, Pfam, KOG/COG, Swiss-Prot, KEGG, and GO [61,62,63,64,65], providing a comprehensive functional context for the gene expression data. This dual-step alignment process was critical for accurately dissecting the host–pathogen interaction at the transcriptomic level.

### 4.6. Analysis of Differentially Expressed Genes

Differential expression analysis was conducted using the DESeq2 (v1.24.0) package in R, which enabled the identification of genes with significant expression changes between the experimental groups [66]. We established stringent criteria to determine DEGs: *p*-value of less than 0.05 for statistical significance, and an absolute log2 fold change (|log2FC|) of 1 or greater to ensure biological relevance. Only genes meeting both thresholds were considered DEGs.

For the quantification of gene expression levels, we utilized RSEM (v1.3.3) to calculate the counts, which were then normalized to transcripts per million (TPM). This normalization accounts for gene length and sequencing depth differences, allowing for accurate cross-sample transcript abundance comparisons.

### 4.7. Quantitative RT-PCR Analysis

To validate the reliability of our transcriptomic findings, we performed quantitative real-time PCR (qRT-PCR) on a subset of six genes from each comparison group (Asn vs. Asp, Jsn vs. Jsp, Asn vs. Jsn, Asp vs. Jsp). The 18s RNA gene was employed as an internal reference for normalization in our qRT-PCR analyses. To ensure the stability and suitability of 18s RNA as a reference, we conducted a stability analysis across all experimental conditions, including the four groups from this study and additional conditions from a related study. The cycle threshold (Ct) values of 18s RNA were assessed and are presented in a box plot format (Appendix A), demonstrating consistent expression across the different conditions, thereby confirming its stability as a reference gene.

Primer sequences for both target and reference genes were crafted using Primer Premier 5 software, with the sequences detailed in Table 3, Table 4, Table 5 and Table 6. The qRT-PCR was carried out in a 20 μL reaction volume that included 2× Power SYBR Green PCR Master Mix, 1 μL of forward and reverse primers, and 1 μL of cDNA. The amplification was initiated with a 5-min denaturation at 95 °C, followed by 40 cycles of denaturation at 95 °C for 10 s and annealing at 57 °C for 20 s. To ensure consistency, each qRT-PCR reaction was performed in triplicate. Appendix A further enumerates the characteristics of the amplified products, such as size and annealing temperatures.

### 4.8. Metabolite Extraction and Liquid LC-MS Analysis

Samples were collected in 2 mL centrifuge tubes and subsequently ground. Metabolites were extracted by adding 400 μL of 80% methanol extraction solution. The sample solution was then ground for 6 min at −10 °C and 50 Hz using a cold tissue grinder. Next, low-temperature ultrasound extraction was performed for 30 min at 5 °C and 40 kHz. Following extraction, the samples were left to stand at −20 °C for 30 min, followed by centrifugation at 4 °C and 13,000× *g* for 15 min. The resulting supernatant was collected for LC-MS analysis. To assess the repeatability of the entire analysis process, a quality control (QC) sample was included every four samples.

LC-MS analysis was conducted using the UHPLC-Q Exactive system manufactured by Thermo Fisher Scientific.

### 4.9. Differential Metabolites Identification

The raw metabolomic data were processed using Progenesis QI (v2.3) software from Waters Corporation, Milford, USA. After importing the raw data, the MS and MSMS spectral information were matched with the metabolic databases HMDB and Metlin. Principal component analysis (PCA) and orthogonal partial Least Squares Discriminant Analysis (OPLS-DA) were performed on the preprocessed matrix file using Ropls (v1.6.2). The stability of the model was assessed using seven rounds of cross-validation.

Differential metabolites were selected based on the OPLS-DA model. Metabolites with a VIP score greater than 1 and a *p*-value less than 0.05 were considered SDMs. The KEGG database was used for metabolic pathway annotation of the SDMs. Pathway enrichment analysis was performed using Scipy.stats (v1.0.0), and Fisher’s exact test was employed to identify the most relevant biological pathways associated with the experimental treatments.

### 4.10. Correlation Analysis of the Transcriptome and Metabolome

To investigate the correlation between the DEGs and SDMs associated with shrimp infection by EHP, we adopted an integrated KEGG pathway enrichment approach. This method involves the enrichment analysis of selected sets of DEGs and SDMs, utilizing the hypergeometric distribution algorithm to identify pathways significantly enriched with genes and metabolites from our datasets. We applied the Benjamini–Hochberg (BH) method to correct for multiple hypothesis testing, with a threshold for significance set at a *p*-value of less than 0.05. Pathways meeting this criterion were considered significantly enriched and indicative of a concerted biological response to EHP infection. This integrative analysis allows for a more robust interpretation of the interplay between the transcriptome and metabolome during the infection process, providing insights into the coordinated changes in gene expression and metabolite abundance within key biological pathways.

### 4.11. Statistical Analysis

Statistical analysis was performed using SPSS 18.0 software. Data are expressed as mean ± standard deviation (SD). Significant differences (copies of EHP in the hepatopancreas) between samples were analyzed by *t*-test with a significance level of 0.05.

## 5. Conclusions

Our study delineates the complex interplay between shrimp and EHP at different infection stages, characterized by marked transcriptional and metabolic shifts. Initially, the immune system ramps up, as shown by the 2444 DEGs and 374 selectivity SDMs, yet some immune genes are paradoxically down-regulated—potentially a tactic by EHP to establish infection. As the disease advances, we see signs of the host immune system adapting, indicative of a longer-term adjustment to the pathogen’s presence. The consistent alteration of energy metabolism pathways, intensifying over time, highlights EHP’s reliance on the host’s energy supply. The late-stage infection also profoundly affects lipid metabolism, likely due to EHP’s proliferation and associated hepatopancreatic damage. Notably, the activation of antioxidant and detoxification pathways later in the infection points to the host’s defense against EHP-induced oxidative stress, with the glutathione system playing a key role in this protective response. The modulation of polyamine metabolism, intertwined with the glutathione pathway, emerges as another critical factor in the host–pathogen dynamic, potentially aiding EHP’s growth. In summary, our findings illuminate the nuanced host–pathogen dynamics in EHP infection, providing a window into the pathogen’s exploitation strategies and the host’s adaptive mechanisms, and thereby contributing to our understanding of EHP pathogenicity.

## Figures and Tables

**Figure 1 ijms-24-16738-f001:**
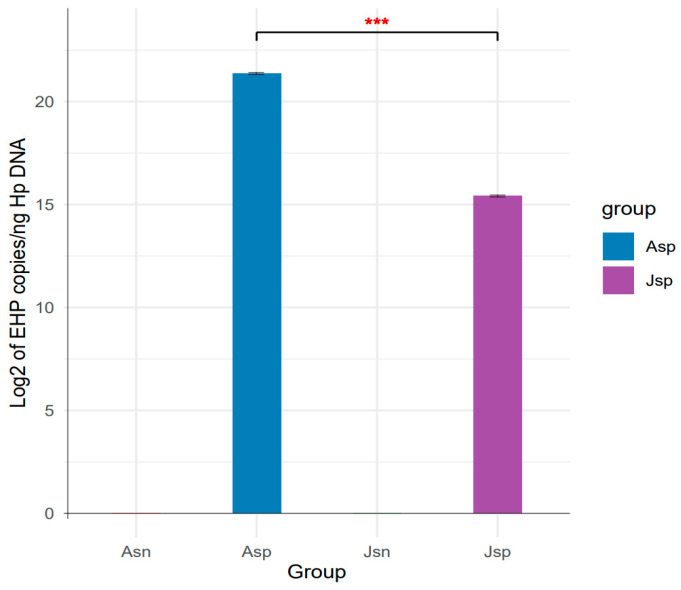
The bar chart shows the copy number of EHP in the hepatopancreas tissue in the four groups of Asn, Asp, Jsn, and Jsp. According to one-way analysis of variance, the Asp group had a significantly higher copy number than the Jsp group (*** *p*-value < 0.001).

**Figure 2 ijms-24-16738-f002:**
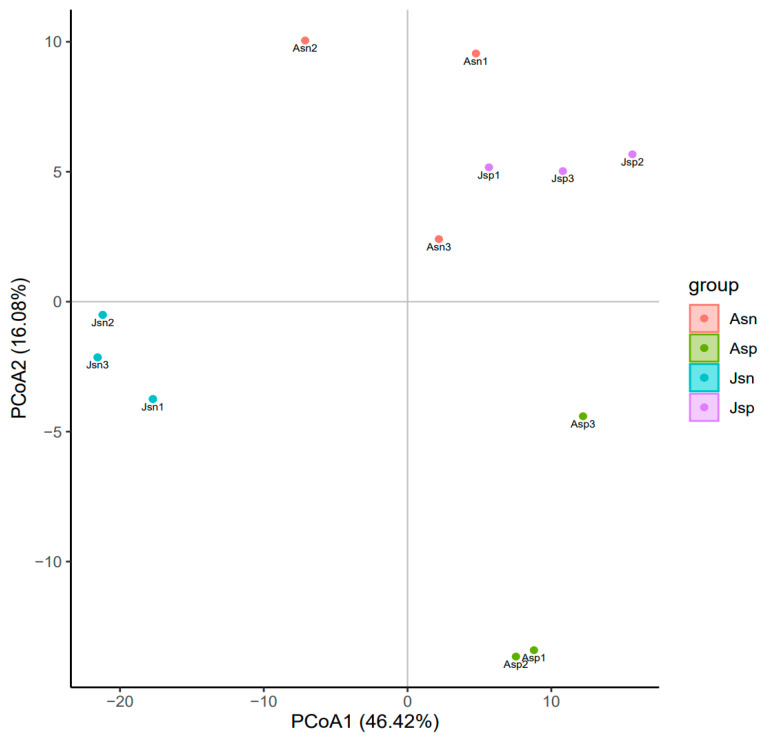
Principal coordinate analysis (PCoA) of transcriptome profiles from healthy and EHP-infected shrimp at various developmental stages.

**Figure 3 ijms-24-16738-f003:**
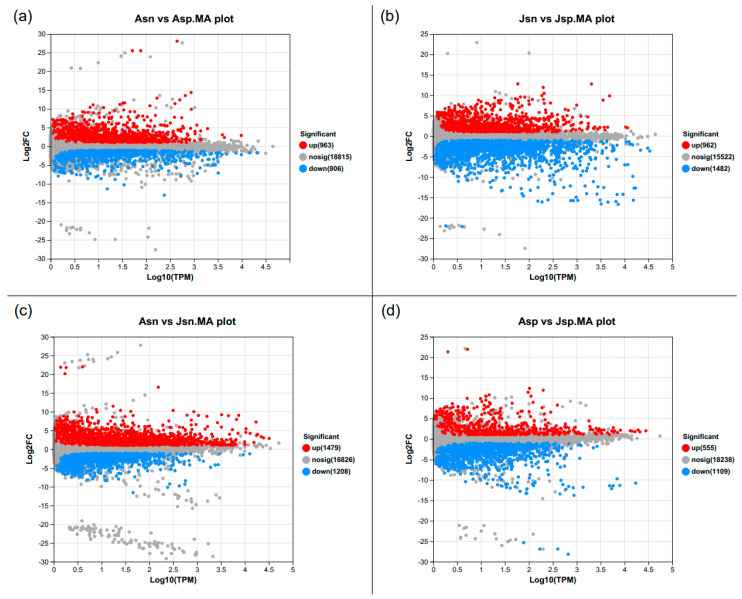
Results of the identification and analysis of DEGs. (**a**–**d**) MA plot of DEGs distribution trends. The horizontal axis is the log10 (TPM) value, the vertical axis is the log2 fold change value. Each dot stands for a gene. Red dots indicate up-regulated DEGs, blue dots are down-regulated DEGs, and gray dots stand for the genes with no significant difference.

**Figure 4 ijms-24-16738-f004:**
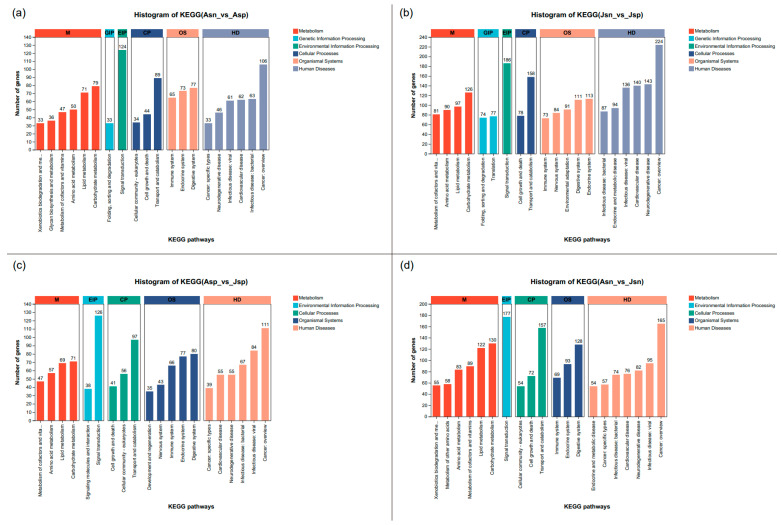
KEGG functional annotation classification of DEGs in the hepatopancreas of EHP-infected and uninfected *L. vannamei*. (**a**): Asn vs. Asp, (**b**): Jsn vs. Jsp, (**c**): Asp vs. Jsp, (**d**): Asn vs. Jsn. (Note: The *x*-axis represents the KEGG pathway category, and the *y*-axis represents the number of unigenes.)

**Figure 5 ijms-24-16738-f005:**
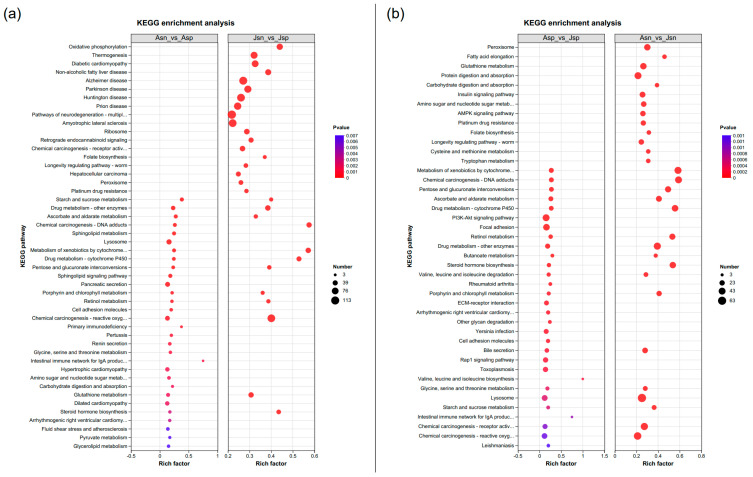
Distribution of KEGG pathway enrichment. (**a**): Asn vs. Asp and Jsn vs. Jsp, (**b**): Asn vs. Jsn and Asp vs. Jsp. The *x*-axis indicates the enrichment factor of the corresponding pathway, and the *y*-axis indicates the KEGG pathway name. Different colors of the dots indicate padj, and the number of DEGs in each pathway is indicated by the size of the dot.

**Figure 6 ijms-24-16738-f006:**
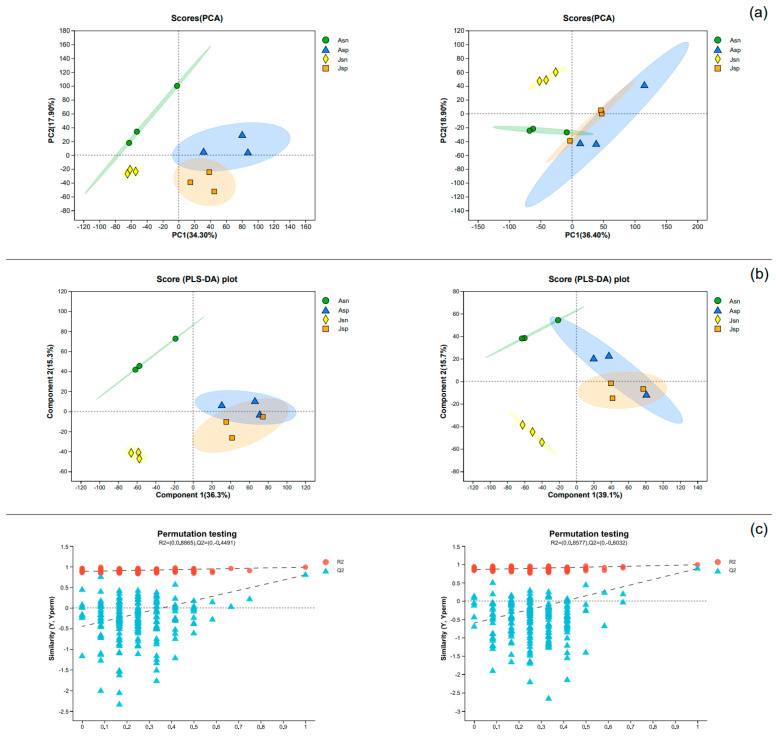
Quality analysis plots of metabolomic data from the hepatopancreas of South American white shrimp at different time periods after EHP infection. The left side is positive ion mode and the right side is negative ion mode. (**a**) Table of PCA principal component analysis in positive and negative ion mode. where the R2X(cum) parameters for the first and second principal components of the positive ion mode are 0.343 and 0.522, respectively, and the R2X(cum) for the first and second principal components of the negative ion mode are 0.364 and 0.554, respectively. (**b**) Partial Least Squares Discriminant Analysis (PLS-DA) score plot for positive and negative ion modes. (**c**) The model validation results of PLS-DA were performed for a total of 200 reciprocal tests of PLS-DA.

**Figure 7 ijms-24-16738-f007:**
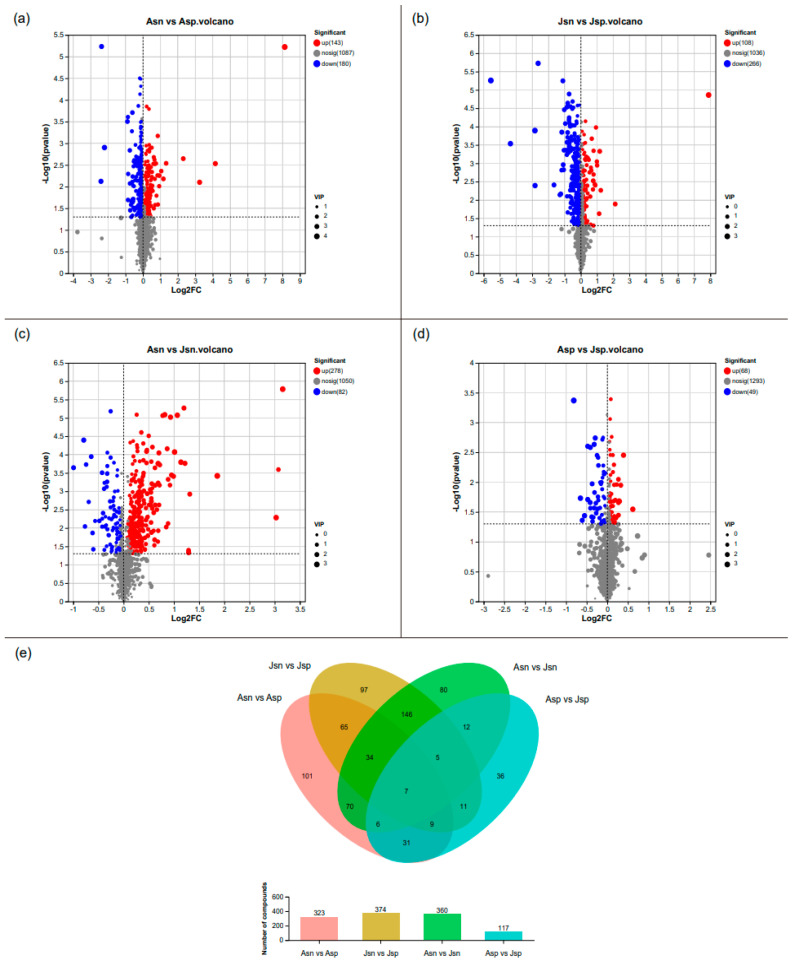
Identification of SDMs with Venn diagram analysis. (**a**–**d**) Volcano plots for the four groups in positive and negative ion mode. The horizontal axis is a log2 fold change value, the vertical axis is −log10 (*p*-value). Each dot represents a metabolite. Red indicates up-regulation, blue indicates down-regulation, and gray indicates no significant difference. (**e**) Venn diagrams of SDMs for four groups. The bar chart indicates the number of SDMs in each group. The different colors indicate four different groups.

**Figure 8 ijms-24-16738-f008:**
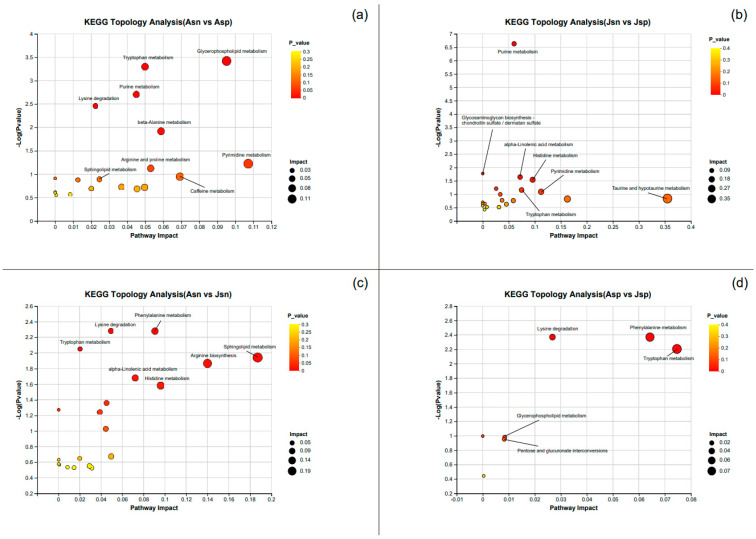
KEGG topology analysis bubble diagram. (**a**): Asn vs. Asp, (**b**): Jsn vs. Jsp, (**c**): Asn vs. Jsn, (**d**): Asp vs. Jsp. The *x*-axis represents the pathway impact, and the *y*-axis represents the pathway enrichment. Larger sizes and redder colors represent greater pathway enrichment and higher pathway impact values, respectively.

**Figure 9 ijms-24-16738-f009:**
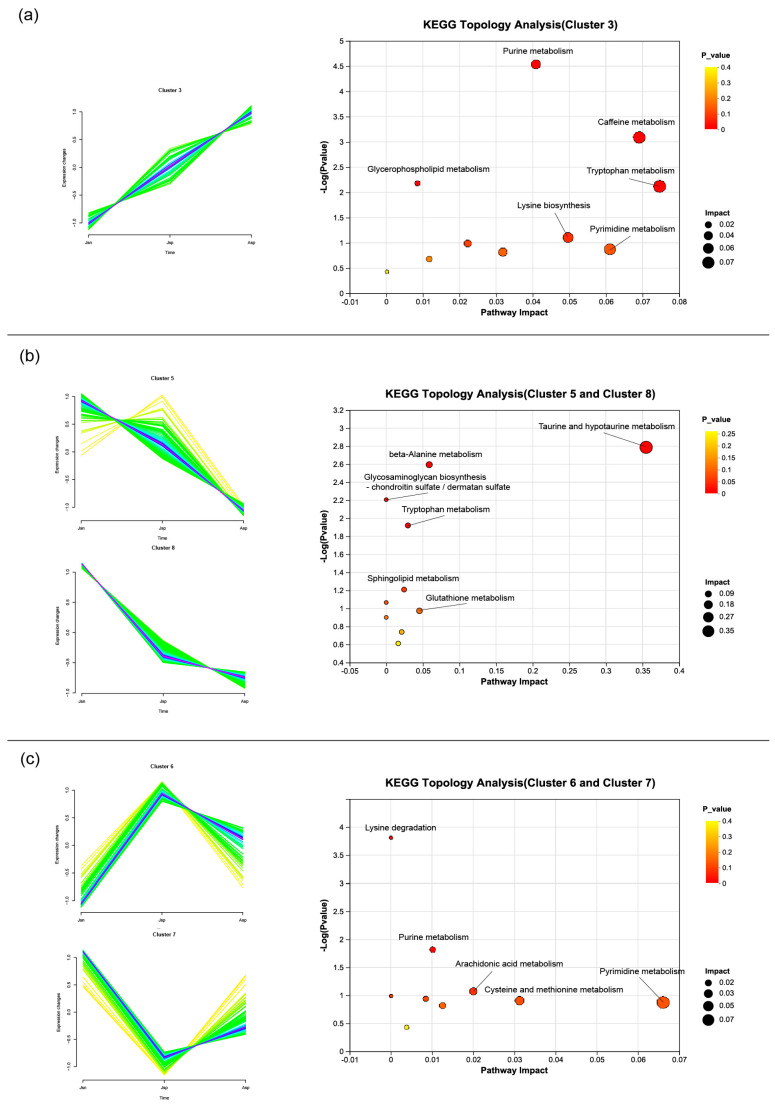
Muffz cluster analysis of SDM from different shrimp groups. Jsn, Jsp and Asp on the horizontal axis represent healthy juvenile shrimp, EHP-infected juvenile shrimp, and EHP-infected adult shrimp. The vertical axis represents the change in abundance of metabolites in each cluster. The left panel categorizes SDMs into three clusters based on their abundance trends: (**a**): This cluster represents differential metabolites with increasing abundance from healthy juvenile shrimp to chronically infected adults, (**b**): This cluster represents differential metabolites that decrease in abundance from healthy juvenile shrimp to chronically infected adults, (**c**): This cluster represents the abundance of these metabolites is altered during the early stage of infection and then recovers to normal levels during the later stage of infection. The right panel shows the KEGG topology analysis, linking these trends to specific metabolic pathways affected by EHP infection.

**Figure 10 ijms-24-16738-f010:**
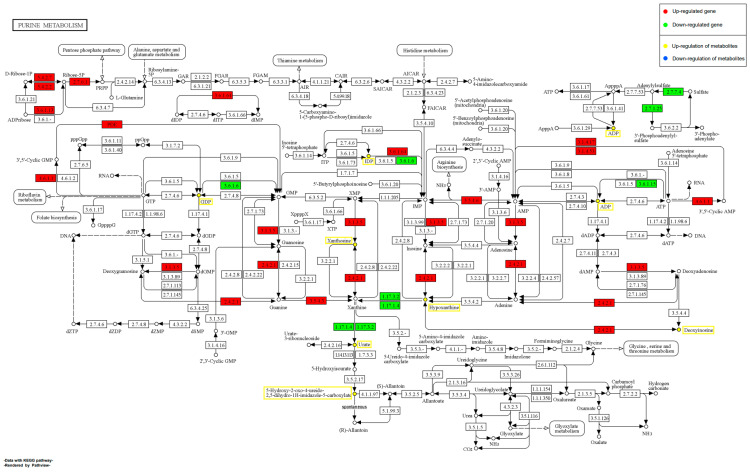
Schematic representation of the pathway of purine metabolism by EHP in the early stages of infection. Red boxes represent up-regulated DEGs, green boxes represent down-regulated DEGs, metabolites in yellow boxes are up-regulated SDMs, and metabolites in blue boxes are down-regulated SDMs.

**Figure 11 ijms-24-16738-f011:**
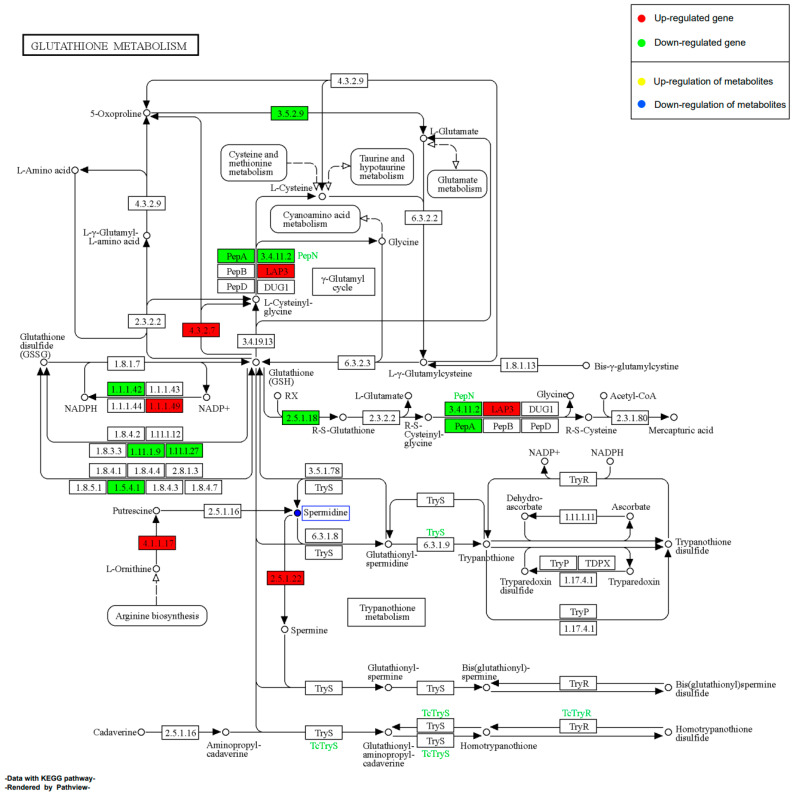
Schematic representation of the pathway of glutathione metabolism by EHP in the early stages of infection. Red boxes represent up-regulated DEGs, green boxes represent down-regulated DEGs, metabolites in yellow boxes are up-regulated SDMs, and metabolites in blue boxes are down-regulated SDMs.

**Figure 12 ijms-24-16738-f012:**
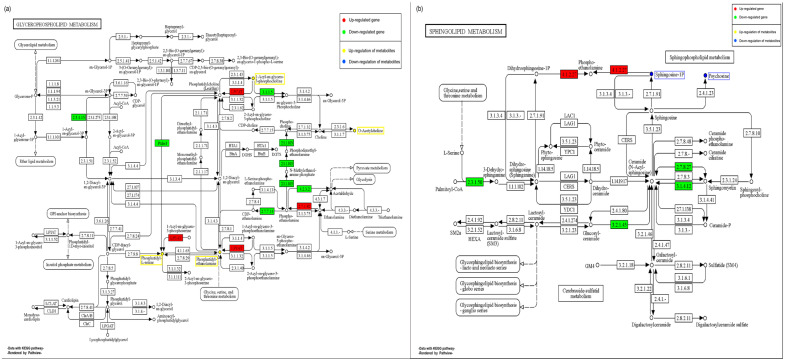
Schematic diagram of the glycerophospholipid, sphingolipid metabolic pathway during the late stages of EHP infection. (**a**): The glycerophospholipid metabolic pathway is shown on the left, (**b**): The sphingolipid metabolic pathway is shown on the right. Red boxes represent up-regulated DEGs, green boxes represent down-regulated DEGs, metabolites in yellow boxes are up-regulated SDMs, and metabolites in blue boxes are down-regulated SDMs.

**Figure 13 ijms-24-16738-f013:**
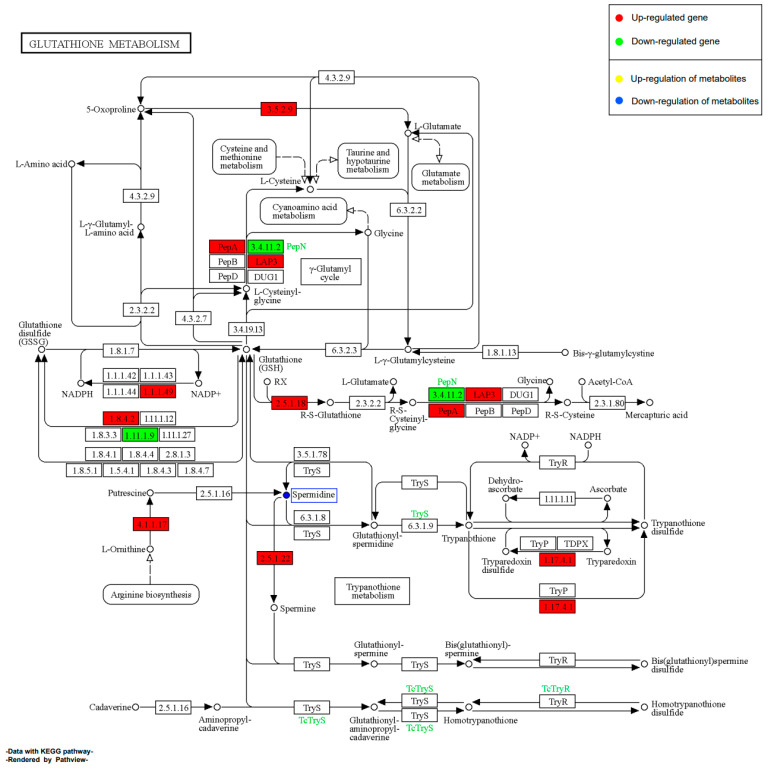
Schematic representation of the pathway of glutathione metabolism by EHP in the late stages of infection. Red boxes represent up-regulated DEGs, green boxes represent down-regulated DEGs, metabolites in yellow boxes are up-regulated SDMs, and metabolites in blue boxes are down-regulated SDMs.

**Table 1 ijms-24-16738-t001:** Summary of the sequencing data of different groups.

Sample	Raw Reads	Clean Reads	Error Rate (%)	Q30 (%)	GC Content (%)
Asn1	60,517,026	58,658,178	0.0254	93.89	47.89
Asn2	50,392,962	48,311,954	0.0252	93.99	48.49
Asn3	46,760,842	45,547,650	0.0251	94.14	48.49
Asp1	73,211,792	71,633,060	0.0249	94.31	49.49
Asp2	51,861,550	50,571,484	0.0251	94.17	48.49
Asp3	44,573,798	43,024,636	0.0255	93.78	48.93
Jsn1	42,488,742	40,402,838	0.0257	93.48	43.21
Jsn2	44,758,988	43,049,092	0.0253	93.94	48.68
Jsn3	42,868,272	40,989,324	0.0255	93.72	44.95
Jsp1	42,872,192	41,092,782	0.0255	93.69	42.85
Jsp2	43,511,964	41,948,760	0.0256	93.62	45.78
Jsp3	42,276,170	41,297,560	0.0256	93.61	44.83

Error rate (%): the average error rate of sequencing bases corresponding to quality control data, generally below 0.1%.

**Table 2 ijms-24-16738-t002:** An annotated summary of genes expressed after EHP infection in *Litopenaeus vannamei*.

	Expre Gene Number	Percentage (%)
GO	13,406	61.37
KEGG	11,612	53.16
COG	15,150	69.36
NR	21,135	96.75
Swiss-Prot	14,008	64.13
Pfam	14,978	68.57
Total	21,844	100

**Table 3 ijms-24-16738-t003:** Sequences of primers used in this study (Asn vs. Asp).

Gene Id	Primer Name	Sequence (5′–3′)
LOC113812219	CLECG4-R	TTGTCGCTGACGCCTTCC
	CLECG4-F	GATGCTGGAGGACTTGAGGAG
LOC113817723	AMY1-R	CTCGGGCTGATGATGTTGAAG
	AMY1-F	CGTGATGTCGTCGTACTACTG
LOC113817801	RPN2-R	GGCTTGGTCAGGCTTAATTGG
	RPN2-F	GATGAAGTTGATGGGCGGATG
LOC113818949	MAN-R	GGTCGCTTCGGTCCTTCAG
	MAN-F	GCCTACGAGCACGGATACAG
LOC113805243	LvPLP-R	CTCCGCTGTGGGTCTTGC
	LvPLP-F	TGTCGCCAGTGTTCGTTCG
LOC113826472	CLCA1-R	ATGACGATGCCGACGATGG
	CLCA1-F	ATGAAGTGAGGGAGATGGTGTC

**Table 4 ijms-24-16738-t004:** Sequences of primers used in this study (Jsn vs. Jsp).

Gene Id	Primer Name	Sequence (5′–3′)
LOC113809619	cenE4-R	TTGGTCAGCGAAGTCGTAGAG
	cenE4-F	CACCCAGGCACGGAGTTG
LOC113813736	RPL13a-R	TTGAAGGAGCACGCTGATGG
	RPL13a-F	CCTTGTGTTCTCGGGCATTTTC
LOC113829084	MGST3-R	ACCATATAGCACCACCAACGG
	MGST3-F	CCACCAGAACACACTTGAGAAC
LOC113818949	MAN-R	GGTCGCTTCGGTCCTTCAG
	MAN-F	GCCTACGAGCACGGATACAG
LOC113823620	HcC-R	GGGAGCAGGAATCGGTTAGG
	HcC-F	TCAGAATCAGCAGTCACAGTCC
LOC113812602	mtATPase-R	AATGATGGCTGGGTCAACTTTC
	mtATPase-F	GCTGCTGTAAGAGGAGAGGTC

**Table 5 ijms-24-16738-t005:** Sequences of primers used in this study (Asn vs. Jsn).

Gene Id	Primer Name	Sequence (5′–3′)
LOC113817185	CNDP-R	AGCCAGTAGTTGTCGGAGATG
	CNDP-F	TCTTTGAGGGAATGGAGGAGTC
LOC113806448	HBPs-R	TTGAATGGTGCGTCGTATGC
	HBPs-F	GAGGAAGTTCGGTGGATATGC
LOC113808797	CP4-R	TCTCCGTCTCGCCGTAGC
	CP4-F	CCAGCCGTCCTTCCAGTTC
LOC113829084	MGST3-R	ACCATATAGCACCACCAACGG
	MGST3-F	CCACCAGAACACACTTGAGAAC
LOC113817723	AMY1-R	CTCGGGCTGATGATGTTGAAG
	AMY1-F	CGTGATGTCGTCGTACTACTG
LOC113827622	DPP1-R	GCAACTGGCGTCCGTATCG
	DPP1-F	GTCGTGCTCCGTGCTGTC

**Table 6 ijms-24-16738-t006:** Sequences of primers used in this study (Asp vs. Jsp).

Gene Id	Primer Name	Sequence (5′–3′)
LOC113829084	MGST3-R	ACCATATAGCACCACCAACGG
	MGST3-F	CCACCAGAACACACTTGAGAAC
LOC113817723	AMY1-R	CTCGGGCTGATGATGTTGAAG
	AMY1-F	CGTGATGTCGTCGTACTACTG
LOC113827551	NEP1-R	GCCACCATAGCATACTTGAAGG
	NEP1-F	ACCACAAACCAGCGGACAC
LOC113827755	TECR-R	TTGCCCACACTGCCATCTG
	TECR-F	CTCACGCTGCTCTTCAATGC
LOC113803442	NCX1-R	CGCCTTCTGAACCTGGATTAAC
	NCX1-F	TGGACCATGAAGACGAGCAC
LOC113810457	Inx2-R	CGCTCCTGTAGTGGCTCTTC
	Inx2-F	TGGCTGGTGCTGCTAATGAC

## Data Availability

The data reported in this paper have been deposited in the NCBI, (https://dataview.ncbi.nlm.nih.gov/object/PRJNA1014601 (accessed on 9 September 2023)).

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
