# Peer review of "Dynamic Interplay of Metabolic and Transcriptional Responses in Shrimp during Early and Late Infection Stages of Enterocytozoon hepatopenaei (EHP)"

_ijms, 2023, doi:10.3390/ijms242316738_

Round 1
Reviewer 1 Report
Comments and Suggestions for Authors
The authors presented a comprehensive study of host-parasite relationship of shrimp Litopenaeus vannamei and a microsporidian Enterocytozoon hepatopenaei (EHP), based on transcriptomic and metabolomic approach. Changes in metabolic and immune related gene sets are presented and discussed, separately for early (juveniles) and later stages (adults). The authors stress the importance of rich dataset in elucidating this host-parasite relationship, obtained through sequential sampling at different time points. However, there are many ambiguities throughout the text that need to be corrected and overall manuscript made more understandable, clear, and concise.
1. Host species name: mostly it is Litopenaeus vannamei, however sometimes you use P. vannamei (like line 108). Please consolidate.
2. Design: the authors refer to the study as timing analyses, time-course analyses, or similar. However, this isn't really time-course analyses as samples were collected at two different time-points from the same ponds. It is unknown when the individual shrimp were infected and the actual age of infection. These are not true sequential sampling points as it would be if this was a controlled experiment. You may talk only about infection in juvenile and adult shrimps (according to their size?). This needs to be acknowledged and explained in the manuscript and corrected throughout. This also makes Mufzz analyses inappropriate.
Furthermore, it is not clear who are adults and who juveniles (size, when were they sampled, where)?
3. Sample comparisons: it states for example:
‘An analysis of differentially expressed genes (DEGs) in Asn vs Asp, Jsn vs Jsp, Asn vs Jsn, and Asp vs Jsp groups was conducted utilizing COG functional annotation…..’(line 143-144)
The authors also use this notation throughout the manuscript.
Does this mean Log2FC were formed by dividing/subtracting Asn/Asp? If Asn are EHP-negative adult shrimp, that means that what we observe in results as downregulation in Asn is actually upregulation in Asp (EHP-positive adult shrimp)?
The confusion becomes apparent in the description of figures 12 and 14.
For figure 12, the authors state (lines 329-330): ‘Among these, the majority of DEGs were significantly downregulated,…..’
However, in the figure 12 most genes are upregulated. Same goes for figure 14, only in reverse.
This could be true if the analyses were done by putting the non-infected state in the numerator, however this is confusing way of presenting analyses, and all the text and figures need to be corrected to clearly point to the same thing. When a statement is made that the upregulation is observed in infected state, the figures need to display this accordingly.
Also, experimental groups are not explained until Materials and methods. This needs to be introduced into the text earlier and mentioned in every figure/table caption.
4. The authors need to make it more clear which differential comparisons are considered specific for infection in juveniles and infection in adults. It is not clear why authors compared for instance Jsn vs Asn – this should be normal development, but it is not properly contrasted in the analyses nor results against EHP. Many of the pathways mentioned as specific for EHP, also appear here.
5. The level of details provided in Materials and Methods is insufficient.
The protocols used for TaqMan probe real-time PCR for the determination of the pathogen load and qPCR used for verification of RNAseq data need to be provided in more detail. All primers also need to be accompanied with product size, annealing temperature and efficiency.
RNAseq lacks the description of sequencing platform.
Another concern is the de novo assembly of host transcriptome for the RNAseq analyses.
Enterocytozoon hepatopenaei, being single-celled eukaryotic microsporidian, was also unavoidably represented in sequencing libraries. The authors do not mention how were these sequences filtered out and if they were filtered out at all.
In lines 110-111 the authors state: ‘Sequencing on the Illumina platform yielded 110.15 Gb of high-quality clean data, with alignment ratios ranging from 78.37% to 91.91% to the NCBI reference genome.’
However, none of this is mentioned in the Materials and methods and what was the purpose.
The authors also need to provide more details on gene enrichment analyses.
6. Reporting of results is insufficient.
The authors have focused on analyzing DEGs through their categorization in different functional sets, like COG, KEGG, GO, however they do not provide a full list of identified DEGs. This should be provided in the supplementary material, along with full results of enrichment analyses (stats for all pathways tested).
This becomes very important especially in the discussion when the authors are referring to immunity related genes. Here for the first time, they mention genes like Ftz-F1 and SEPs, not mentioned in the results, nor in any provided tables or figures.
From the provided material, I do not find evidence of the perturbation of the immune system during EHP infection in shrimp, and this forms a large part of the abstract, conclusions and the discussion, however barely mentioned in the results.
The part pertaining to genes related to metabolism and metabolites is much better presented and discussed.
7. Many of the tables and figures are redundant and or not readable due to small resolution or text. This needs to be corrected.
Table 1. – not necessary (all is explained in lines 96 – 98)
Figure 2. (line 104) – this actually figure 1
Figure 2. (line 134) adjust comparisons according to comment 3
Figure 3. Needs to be made more readable. Also, a large part of DEGs is of unknown function. Please also mention this in the text.
Figure 4. Not necessary, or place in supplement. GO categories are too general and can relate to any genes. This is not proof of alterations in cellular structures and organization due to the influence of EHP (lines 159-160)
Figure 5. Needs to be made more readable (bigger text). Please explain what Rich factor is.
Figure 6. Not necessary or place in supplement. This is just technical validation of RNAseq results. One could easily describe it as correlation between two datasets and the statistical significance.
Figure 7. very nice figure, nicely shows sample relationships for metabolomic data.
Please also provide PCA for RNAseq data. It would be nice to see if a similar pattern is observed.
Figure 8. adjust comparisons according to comment 3.
Lines 245 -247 this is self-explanatory, no need to mention it in figure caption. Instead, please describe experimental groups.
Figure 9 adjust comparisons according to comment 3.
Figures 11 – 14 too many KEGG maps. Choose which are most important, the rest can be placed in supplemental material.
8. Not sure where the results of Correlation analysis of the transcriptome and metabolome (line 739) are?
Thank you for submitting and providing all the sequences under PRJNA1014601 and making them available during the review process.
Author Response
Please see the attachment, thanks.

Reviewer 2 Report
Comments and Suggestions for Authors
The authors investigated the molecular interactions between Enterocytozoon hepatopenaei (EHP) and its host, the Pacific white shrimp, through an integrated analysis comprising transcriptomics and metabolomics, during the early and the late infection phases. Results are promising indicating distinct responses between both infection stages, with a common significant disruption of the polyamine metabolism. Results and discussion sections are well interpreted and complete. My only comments go to the M&M section, specifically the lack of description regarding the pre-processing and quality control analyses of short reads, and the use of a single reference gene in RT-PCR. A single reference gene in RT-PCR analyses is not accepted anymore by the MIQE guidelines, the authors should at least include a stability analysis.
Author Response
Please see the attachment, thanks

Round 2
Reviewer 1 Report
Comments and Suggestions for Authors
Dear Authors,
Thank you for carefully considering all points raised and providing constructive and comprehensive revisions throughout the manuscript. I believe most major concerns have been resolved, especially the ones about cleaning RNAseq reads from EHP sequences and the study design referred to as temporal analyses. I agree, cluster approach is a better solution.
I do not see much improvement in figure resolution though, but this may also be due to embedding in the document which will hopefully be resolved at final editing step.
I still have couple of major concerns I would like to point out that would greatly improve the manuscript:
- The full list of DEGs is still not provided. Several chosen are in supplementary material, however many mentioned in the discussion are not (COX, hexokinase, PGM), also including Ftz-F1 and SEPs. They are still missing form results. Either remove these genes from discussion, or provide a complete list of DEGs, with gene enrichment testing results. An excel sheet can also be a supplement.
The role of immune genes is over-emphasized. Even with provided gene data, there are a handful of selected immunity genes, while all other metabolic analyses are supported by pathway enrichment data and many more DEGs. I do not see evidence for statements made in lines 485-490; 495-499; 505-513; 526 -527; 528-533; 547
They also seem out of place with the rest of the discussion in context they are placed in. They need to be all collected in one shorter paragraph together with lines 515-521, and the role of immunity downplayed in the conclusions and abstract. All that is mentioned in the abstract is too general (lines 16-18) and needs to be rephrased.
- Although greatly improved and clearer, the manuscript is quite long and cumbersome to read. To increase its impact and readability, I suggest removing all that is GO and COG analyses. These are only showing the distribution of DEGs in various functional groups, but since you have enough annotation and power for an enrichment analysis, as was done, these become redundant and carry no added value. They were not discussed either. You focused on KEGG for interpretation, which is valid and provided strong results.
I also suggest removing Muffz analyses. It is only done for metabolomic data, not for transcriptomics, so here they are not synchronized. Additionally, although there is some added value to it, it essentially identified the same pathways as KEGG enrichment analyses. Plus, this essentially is not a time-cource experiment.
Other minor corrections:
Line 41 It is stated 70% in reference 1
Lines 69-73 can be omitted. They are too general and common knowledge.
Line 82 we gained
Line 86 Our primary goal was to
Line 87 transcriptomics and metabolomics approach
Line 88 we explored
Lines 114-115 ensuring its suitability for transcriptome analyses
Line 124 Table 2 Expr? Total anno?
Please add N50 of the transcriptome assembly and few other quality metrics
Lines 157 – 158 Delete the sentence. No, the moment there is interaction, individual impacts become less important
Lines 168 – 170 delete
Line 173 indicated
Line 174 were significantly impacted
Line 207 delete: Analogous to the GO functional enrichment analysis,
Line 210 delete: In our approach
Lines 233 -234 according to this statement, you need to provide this information for all DEGs
Lines 245 – 246 deleted the sentence.
Line 251 please specify in the section title that this is for metabolomics data
Lines 300 – 302 delete.
Lines 307 – 312 is it not that Figure 9a is late response and 9b early response?
Line 398 Figure 11 DEGs are represented by EC numbers. This is difficult to follow in respect to discussion. Please annotate with at least most important gene names as in Figure 12
The same for Figure 13
Line 453 EHP instead of Microsporidia (as in other figure captions)
Line 457 EHP instead of microsporidia
Line 472 same as in 453 and 457
Line 605 disruption of hepatopancreas function
Line 757 The detailed protocol is not provided in Table S2, only the list of primers
Line 765 wrong subsection title?
Line 802 adjusted p-value?
Lines 806 – 812 it is not described how was enrichment analyses conducted for DEGs
Line 826 please provide primer concentration.
Line 828 please describe how were qPCR data further analysed
Lines 830 – 833 Product size, efficiency, and annealing temperature are still not provided. Please place all these tables in the supplement
Line 871 Please state for which data were these analyses conducted. Was the normality and heteroscedasticity tested and if any transformations to the data were applied.
Line 876 please shorten the conclusion; this is too long. Actually, it should be more in the style of lines 715 – 720.
Author Response
Dear Reviewer 1,
We greatly appreciate your constructive comments and the time you have invested in reviewing our manuscript. We have carefully considered each point and have made the following revisions to address your concerns:
Complete List of DEGs:
We acknowledge the importance of transparency in reporting all DEGs discussed in the manuscript. To this end, we have included a comprehensive list of DEGs, including those previously mentioned in the discussion (COX, hexokinase, PGM, Ftz-F1, and SEPs), in a supplementary Excel sheet. This sheet now accompanies the manuscript and contains full gene enrichment testing results.
Regarding Immune Genes:
We appreciate your guidance on the analysis of immune-related genes. In response, we have enhanced the supplementary materials by adding more detailed visualizations and comprehensive data, particularly focusing on immune-related genes. This augmentation aims to provide a clearer understanding and aligns with your requirements for a more in-depth examination of these aspects.
GO and COG Analysis Removal:
Upon reflection, we agree that the GO and COG analyses do not add significant value to our findings in light of the detailed KEGG pathway enrichment analysis. As such, we have removed these sections from the manuscript to streamline the content and enhance its readability.
On the Use of Mufzz Analysis:
We acknowledge the concern regarding the inclusion of Muffz analysis in our manuscript. While we have previously employed correlation clustering analysis for transcriptome data, in this study, we also include the analysis of certain metabolites, like taurine, underscoring their significance in our discussion. We recognize that relying solely on KEGG pathway enrichment might not capture the complete picture, especially in the context of topological analysis. We understand that our experimental design may not perfectly align with timing analysis, and we appreciate your patience and understanding of this aspect of our study.
Minor Corrections:
We are grateful for your meticulous attention to the details of our manuscript. All the minor corrections you pointed out have been carefully addressed, and the necessary revisions have been made throughout the manuscript, including the abstract and conclusion sections. These changes have greatly enhanced the clarity and accuracy of our work.
Manuscript Readability and Length:
In addition to the above, we have taken steps to shorten the manuscript and improve its flow. We have removed general knowledge statements and redundant sentences, tightened the conclusions, and ensured that each section is directly relevant to our study's objectives and findings.
We believe these revisions have significantly improved the manuscript and hope that it now meets your expectations for publication. Thank you once again for your invaluable feedback.
Kind regards,
Hui Shen
